# Plastomes of *Garcinia mangostana* L. and Comparative Analysis with Other *Garcinia* Species

**DOI:** 10.3390/plants12040930

**Published:** 2023-02-17

**Authors:** Ching-Ching Wee, Nor Azlan Nor Muhammad, Vijay Kumar Subbiah, Masanori Arita, Yasukazu Nakamura, Hoe-Han Goh

**Affiliations:** 1Institute of Systems Biology, Universiti Kebangsaan Malaysia, Bangi 43600, Selangor, Malaysia; 2Biotechnology Research Institute, Universiti Malaysia Sabah, Kota Kinabalu 88400, Sabah, Malaysia; 3Department of Informatics, National Institute of Genetics, Mishima 411-8540, Shizuoka, Japan

**Keywords:** *Garcinia*, Manggis, Mesta, phylogenomic analysis, plastome

## Abstract

The two varieties of mangosteen (*Garcinia mangostana* L.) cultivated in Malaysia are known as Manggis and Mesta. The latter is preferred for its flavor, texture, and seedlessness. Here, we report a complete plastome (156,580 bp) of the Mesta variety that was obtained through a hybrid assembly approach using PacBio and Illumina sequencing reads. It encompasses a large single-copy (LSC) region (85,383 bp) and a small single-copy (SSC) region (17,137 bp) that are separated by 27,230 bp of inverted repeat (IR) regions at both ends. The plastome comprises 128 genes, namely, 83 protein-coding genes, 37 tRNA genes, and 8 rRNA genes. The plastome of the Manggis variety (156,582 bp) obtained from reference-guided assembly of Illumina reads was found to be nearly identical to Mesta except for two indels and the presence of a single-nucleotide polymorphism (SNP). Comparative analyses with other publicly available *Garcinia* plastomes, including *G. anomala*, *G. gummi-gutta*, *G. mangostana* var. Thailand, *G. oblongifolia*, *G. paucinervis*, and *G. pedunculata*, found that the gene content, gene order, and gene orientation were highly conserved among the *Garcinia* species. Phylogenomic analysis divided the six *Garcinia* plastomes into three groups, with the Mesta and Manggis varieties clustered closer to *G. anomala*, *G. gummi-gutta*, and *G. oblongifolia*, while the Thailand variety clustered with *G. pedunculata* in another group. These findings serve as future references for the identification of species or varieties and facilitate phylogenomic analysis of lineages from the *Garcinia* genus to better understand their evolutionary history.

## 1. Introduction

Mangosteen (*Garcinia mangostana* L.) is well known as the ‘queen of fruits’ and it is priced for its unique taste and valuable natural compounds. Xanthones, which are abundantly found in the ripe fruit pericarp, have been shown to possess antioxidant, anti-cancer, anti-inflammatory, anti-bacterial, and anti-viral properties [1]. Mangosteen is mainly found in Southeast Asia, particularly in Malaysia, Indonesia, and Thailand [2]. The geographical origin of *G. mangostana* is still under debate. Unlike other flowering plants, *G. mangostana* reproduces apomictically by adventitious embryony in the mother plant without fertilization [3] and produces *Garcinia*-type recalcitrant seeds without embryo [4]. Morphological and phylogenetic analyses have been performed to examine the parental origin of *G. mangostana* and its relationship with other *Garcinia* species on the basis of internal transcribed spacer (ITS) [5,6,7], granule-bound starch synthase (*GBSSI*) [7], *trnS-trnG*, and combination of *trnS-trnG* with *trnD-trnT* [8]. They showed that *G. mangostana* was closely related to *G. malaccensis*, and as such, were postulated to have been derived from the hybridization of *G. hombroniana* and *G. malaccensis* [9]. However, as there was only one mangosteen sample (*G. mangostana* TH3) that showed heterozygosity in the ITS sequence, Nazre proposed *G. mangostana* and *G. malaccensis* to be grouped as one species but different varieties [10]. Mangosteen was suggested to have originated either from the hybrid of different varieties of *G. malaccensis* or the product of agricultural selective breeding retaining only superior female plants of *G. malaccensis* [10].

Nonetheless, several reports showed that molecular markers from the nuclear genome could not provide sufficient information for phylogeny demarcation [11]. This is likely due to recombination events in the plant nuclear genome during reproduction [12]. In contrast, the majority of the plastome is inherited maternally. Hence, a plastome with a slower rate of evolution provides a better resolution in examining species phylogenetic relationships, adaptive evolution, and divergence dating [13,14]. Recently, with the advancement of next-generation sequencing and long-read sequencing technology, complete plastomes have been able to be obtained easily at low costs. A complete plastome of *G. mangostana* of an unspecified variety that originated from Thailand was first reported in 2017 with the accession number KX822787 [15] (herein, denoted as the Thailand variety) and was shown to be closely related to *G. pedunculata* [15]. In Malaysia, two varieties of *G. mangostana* (Manggis and Mesta) were sequenced and deposited to GenBank [16,17,18]. The mitogenome of the Mesta variety was reported recently [19]. However, a complete analysis of the plastomes from these two varieties is yet to be reported.

In this study, we assembled and analyzed the complete mangosteen plastomes of Manggis and Mesta. Furthermore, we performed a comprehensive comparison of all available *Garcinia* plastomes from GenBank and provided an update to a previous comparative analysis [20]. The current comparative study elucidates the structural differences in plastomes for evolutionary inference of the *Garcinia* genus.

## 2. Results

### 2.1. Characterization of the Mesta Plastome

De novo assembly of PacBio subreads data using the CANU assembler and error-correction with Illumina data using the Pilon program produced a total of 7616 contigs with an N50 genome length of 10,212 bp (Appendix A). There was one contig (tig00037541_pilon) with the size of ≈165 kb that showed high similarity with the reference plastome in BLAT analysis. It was a circular contig, as indicated by the dot plot analysis using Gepard [21] (Appendix A). One of the identical overlapping ends (≈9.3 kb) was trimmed, and 18 bases were manually added on the basis of Illumina read correction to obtain the final Mesta plastome size of 156,580 bp. The average coverages of the Mesta plastome with PacBio subreads and Illumina clean reads were 265× and 3751×, respectively (Appendix A).

The Mesta plastome constituted a typical conserved quadripartite structure with two inverted repeat (IR) regions (each 27,030 bp) separating the large single-copy (LSC) region (85,383 bp) from the small single-copy (SSC) region (17,137 bp) (Figure 1). The average GC content of the plastome was 36.2%, while the GC contents in LSC, SSC, and IR regions were 33.6%, 30.2%, and 42.2%, respectively. A total of 128 genes were identified, including 77 unique protein-coding genes with six duplicated genes in IR, 30 unique tRNAs (seven duplicated genes in IR), and 4 rRNAs (four duplicated genes in IR) (Table 1 and Table 2).

### 2.2. Manggis Plastome Assembly

The Manggis Illumina clean reads had a higher mapping rate to the Mesta plastome than the Thailand variety (Appendix A). Hence, the Mesta plastome was used for a reference-guided genome assembly of the Manggis plastome. The complete plastome of Manggis had a genome size of 156,582 bp (Table 1) after manual curation with the same genome features as observed in the Mesta plastome (Figure 1), except for one single-nucleotide polymorphism (SNP) and two indels (Appendix A). The average coverage of the Manggis plastome with the Illumina short reads was 2636× (Appendix A).

### 2.3. Plastome Feature Comparison

In comparison with the plastomes of other *Garcinia* species, the plastome sizes of both Mesta and Manggis varieties (156,580 bp and 156,582 bp, respectively) were larger than *G. gummi-gutta* (156,202 bp) and *G. oblongifolia* (156,577 bp), but smaller than *G. anomala* (156,774 bp), *G. pedunculata* (157,688 bp), *G. paucinervis* (157,702 bp), and the Thailand variety (158,179 bp) (Table 1).

Gene *infA* was found in GenBank for *G. pedunculata* (NC_048983) and *G. anomala* (MW582313), while the gene *rpl32* was found for *G. pedunculata* (NC_048983). However, both genes were not annotated accurately. Multiple sequence alignment of the *infA* and *rpl32* genes (Appendix A) with other species showed that both annotated sequences did not have a conserved region as compared with other species. Hence, these genes were re-annotated and revised for accuracy prior to subsequent comparative analysis.

The number of protein-coding genes (83 CDS) was the same for all the *Garcinia* species. However, the gene *trnH-GUG* was not found in *G. gummi-gutta*. Hence, the total number of genes for *G. gummi-gutta* was 127 compared to 128 genes for other *Garcinia* species (Table 1). The overall GC content (36.1–36.2%) and GC content found in LSC (33.5–33.6%), SSC (30.1–30.3%), and IR (42.1–42.2%) were similar among the *Garcinia* species.

In the plastomes of six *Garcinia* species (including *G. mangostana* var. Manggis, Mesta, and Thailand) used in this comparative analysis, there were a total of 18 single-copy non-redundant plastid genes (*rps16*, *atpF*, *rpoC1*, *ycf3*, *rps12*, *clpP*, *petB*, *petD*, *rpl16*, *rpl2*, *ndhB*, *ndhA*, *trnK-UUU*, *trnG-UCC*, *trnL-UAA*, *trnV-UAC*, *trnI-GAU*, and *trnA-UGC*) containing at least one intron with two introns in *clpP* and *rps12* (Appendix A), which is similar to those generally found in other plants [22]. Gene *clpP* was located in the LSC region. Meanwhile, the 5′ exon of the *rps12* gene was in the LSC region, while the 3′ exon was located in the IR regions, which is commonly observed in plastomes of other species such as *Rhodomyrtus tomentosa*, *Salvia* spp. [23,24], *Ananas comosus* var. *comosus* [22], and ferns [25]. Among the genes, *trnK-UUU* had the longest intron length, which agrees with previous studies [26,27,28].

### 2.4. Codon Usage and Amino Acid Frequency

The total numbers of codon usage in 83 protein-coding genes found in the plastomes was different among the *Garcinia* species, ranging from 26,195 in *G. pedunculata*, 26,216 in Thailand variety, 26,244 in *G. paucinervies*, 26,249 in *G. anomala*, 26,257 in both Manggis and Mesta varieties, 26,265 in *G. gummi-gutta*, to 26,268 in *G. oblongifolia* (refer to ‘total number of codon usage’ in Appendix A). There were several common findings in the codon usage analysis of *Garcinia* plastomes: (1) a total of 20 translated amino acids; (2) the most frequent amino acid was leucine, while the least frequent was cysteine (Appendix A); (3) there were 30 types of codon out of 64 codons with relative synonymous codon usage (RSCU) values >1.0 (ending with either A or U, except for UUG) and 32 types of codons with RSCU < 1.0 (ending with either C or G, except for AUA, CUA, and UGA); (4) both AUG and UGG had RSCU values = 1 (Appendix A). Similar findings were also detected in other plants, such as Euphorbiaceae and *Rhodomyrtus tomentosa* [24,29].

Generally, the start codon is ATG, but there were exceptions for several genes with initiation codons of GTG or ACG due to the RNA editing events during transcription. The first discovery of such an event came from the study of the *rpl2* gene in the maize plastome when the start codon of this gene changed from ACG to ATG during transcription [30]. Other examples include GTG as an initiation codon of the *psbC* gene [31] and ACG for the *ndhD* gene [32] in the tobacco plastome, as well as start codons ACG and GTG for *rpl2* and *rps19* genes, respectively, in the *Oryza sativa* plastome [33]. Similarly, there were three genes (*rpl33*, *rps19*, and *ndhD*) in the plastomes of *Garcinia* species that did not start with ATG. The start codon was GTG for both *rpl33* and *rps19* genes in all the *Garcinia* species, except for *rps19* in *G. anomala*, which started with an ATG. As for *ndhD*, ATG was found in *G. anomala*, while TTG was found in *G. pedunculata*, and ACG was found for all the other *Garcinia* species.

### 2.5. Simple Sequence Repeat (SSR) Analysis

A total of 88 SSRs were identified, including 79 mononucleotide repeats, 7 dinucleotide repeats, and 2 trinucleotide repeats with a total sequence size of 1065 bp and 1067 bp for Mesta and Manggis varieties, respectively. The SSRs identified in the plastomes of both Mesta and Manggis were nearly the same, except Manggis had additional mononucleotides, A (SSR (A)14 and (A)15 at no. 3) and C (SSR (C)12(A)12 and (C)13(A)12 at no. 25), as per Appendix A, respectively, for Mesta and Manggis. There were 14 compound SSRs in both Mesta and Manggis. The most abundant motif found in both Mesta and Manggis varieties was mononucleotide repeats (89.8%), in which mononucleotide T (48.9%) and A (38.6%) constituted the highest portion compared to mononucleotide C (1.1%) and G (1.1%). Among them, only 10 of the mononucleotide repeats were located at the coding regions of *rpoC2*, *rpoC1*, *rpoB*, *rps19*, *ycf2*, and *ycf1* genes. All dinucleotide and trinucleotide repeats were located at the non-coding regions, which are common phenomena in flowering plants [34]. All the SSRs in the plastomes of Mesta and Manggis varieties including their respective locations are listed in Appendix A, respectively.

The total number of SSR varied among the different *Garcinia* species (Figure 2). Mesta and Manggis varieties had the lowest number of total SSR (88) compared to *G. oblongifolia* (106), Thailand variety (105), *G. gummi-gutta* (102), *G. anomala* (96), *G. paucinervis* (94), and *G. pedunculata* (91). Mononucleotide repeats constituted the highest percentage in SSR analysis in this study, agreeing with the previous studies of plastomes from 164 lower and higher plants [34]. The highest dinucleotide repeat was found in *G. oblongifolia* (11), followed by the Thailand variety (8) and Mesta/Manggis varieties (7), while the other *Garcinia* species showed six dinucleotide repeats. The Thailand variety had three trinucleotide repeats, being the highest among all the *Garcinia* species. There was only one trinucleotide repeat found in both *G. gummi-gutta* and *G. paucinervis* compared to two trinucleotide repeats found in Mesta, Manggis, and *G. oblongifolia*. Trinucleotide repeat was not found in *G. anomala* and *G. pedunculata*. Most of the mononucleotide SSR belonged to the A/T repeats, and the same findings were observed in the previous studies [22,24,27].

### 2.6. Long Repeat Analysis

By using the default setting of 50 for maximum computed repeats, different types of long repeats were detected in the plastomes of *Garcinia* species, including forward repeats, reverse repeats, complement repeats, and palindromic repeats (Figure 3). The palindromic repeat was the most common repeat found in *Garcinia* species, followed by the forward repeats, which was also observed in other plants [28]. Mesta and Manggis varieties had the highest number of palindromic repeats of 31, while *G. anomala* had the lowest number of palindromic repeats of 25. The reverse repeat was found in all *Garcinia* species, except *G. gummi-gutta*, which had the highest number of forward repeats (21). *G. paucinervis* had the highest number of reverse repeats (5), followed by three reverse repeats found in the Thailand variety and *G. anomala*, while the other *Garcinia* species only had one reverse repeat. In addition, complement repeat was only found in *G. anomala* (2), Mesta and Manggis varieties (1), and *G. paucinervis* (1).

### 2.7. Contraction and Expansion of the Inverted Repeat Region

The smallest inverted repeat (IR) was found in *G. paucinervis* (26,988 bp), while the largest IR was found in *G. oblongifolia* (27,060 bp) (Figure 4). Genes that can be found at or close to the junctions of IRs were *rps19*, *ndhF*, *ycf1*, and *trnH*. The gene *rps19* was located at the LSC/IRb junction site (JS) and the fragment size located at LSC in all *Garcinia* species was 60 bp, except for the Thailand variety, which was only 8 bp. In addition, the *rps19* gene fragment of the Thailand variety located at the IRb site was 1 bp longer (220 bp) than the *rps19* gene fragment (219 bp) of the other *Garcinia* species at the same location. The *ndhF* gene spanned across the SSC/IRb with 1 bp located at the IRb region for *G. anomala*, *G. gummi-gutta*, Manggis and Mesta varieties, and *G. oblongifolia*. However, it was 1 bp away from the SSC/IRb junction site of both *G. paucinervis* and *G. pedunculata*. Interestingly, the Thailand mangosteen was the only one with *ndhF* gene 4 bp from the SSC/IRb junction. The *ycf1* gene fragment (1421 bp) in the IRa region was the same for all *Garcinia* species, except for *G. pedunculata* and *G. paucinervis* (1419 bp). The size of the *ycf1* fragment in the SSC region ranged from 4204 to 4245 bp. In addition, it was found that *tRNA-trnH* was missing at the IRa/LSC junction of *G. gummi-gutta*. In comparison with *Erythroxylum novogranatense* (Plastome size: 163,937 bp; LSC: 91,383 bp; SSC: 18,138 bp; IR: 27,208 bp), a sister group of *G. mangostana* [15], plastome size, LSC, SSC, and IR regions of *Garcinia* species were much shorter (Figure 4, Table 1).

### 2.8. Comparative Plastome Analysis

Plastome comparison using Mesta as a reference was performed using the mVista online alignment tool (Figure 5). The qualitative comparison among *Garcinia* species showed that (1) IR regions were more conserved (higher % identity) compared with LSC and SSC regions, and (2) coding regions were more conserved (higher % identity) than intergenic regions. This result agreed with previous reports in Plantaginaceae, Rosaceae, and Sapindaceae families [26,35,36]. For LSC, highly divergent intergenic regions include *trnH-psbA*, *trnQ-psbK*, *trnG-trnR*, *atpF-atpH*, *atpH-atpI*, *trnT-psbD*, *ndhC-trnV*, *rbcL-accD*, *psbB-psbT*, and the intergenic region within the *rpl16* gene. As for SSC, the highly divergent regions include *ndhF-trnL* and the intergenic region within the *ndhA* gene. Divergent regions were also found in the coding regions such as *matK*, *rpoC2*, *rpoC1*, *rpoB*, *rbcL*, *accD*, *ycf4*, *cernA*, *petA*, *petD*, *rpoA*, *ndhF*, *ccsA*, *ycf1*, and *ycf2* (Figure 5). In comparison with *Erythroxylum novogranatense*, plastome sequences of *Garcinia* species were conserved within the Clusiaceae family.

Multiple genome (plastome) alignment between 17 species from the order Malpighiales (Appendix A) using Mauve with *E. novogranatense* as a reference detected one inversion of ≈360 bp. It was located between *trnV-UAC* and the *atpE* gene with 15 bp palindromic repeats (ACATCCTATTTCTTT/AAAGAAATAGGATGT) detected at the break point of both sites of inversion. Surprisingly, this inversion was only found in *Garcinia* species (Figure 6) but not in other species of the same order.

### 2.9. Phylogenomic Analysis

A total of 74 protein-coding genes (Appendix A) were used for phylogenomic analysis. Phylogenomic analysis showed that both Manggis and Mesta varieties were grouped together as the CDS sequences were 100% identical, despite some base differences in non-coding regions. Both were grouped under the clade of *Garcinia* species in the Malpighiales order among the three groups of *Garcinia* species (Figure 7). *G. anomala* and *G. gummi-gutta* were closely related and formed one group with the Mesta/Manggis varieties and *G. oblongifolia*. The Thailand variety and *G. pedunculata* formed another group, while *G. paucinervis* formed the third group in the Clusiaceae family.

## 3. Discussion

Plant DNA is rich in plastome (≈5–20%) and hence, an enrichment strategy is not required for sequencing [14]. In this study, a complete mangosteen plastome of Mesta variety (156,580 bp) was successfully obtained from PacBio long reads. Here, the use of long reads for the assembly of the plastid genomes was ideal to obtain longer contigs and to resolve repetitive regions [27,37,38,39]. In addition, Illumina sequencing data were used to correct the random errors within PacBio reads [40]. Furthermore, the polished Mesta plastome allowed for the reference-guided assembly of the plastome from Manggis, which had only Illumina short reads.

Both the Mesta and Manggis plastomes were nearly identical (Figure 1), consisting of a typical conserved quadripartite plastome structure found in most of the land plants [14,41]. Generally, the number of genes encoded in a plastome ranges from 110 to 130 genes [36]. Both Malaysian mangosteen plastomes fall within the range with 128 genes (111 unique genes), consistent with all other *Garcinia* species found in GenBank, except for *G. gummi-gutta*, which lost one *trnH* gene (Table 1). Typically, plastomes have 30–31 tRNAs [42,43], and the loss of tRNA genes is not uncommon. For instance, two hemiparasitic *Taxillus* species lost seven tRNAs, including *tRNA-trnH* [44]. Sometimes, missing tRNAs might be replaced by other types of anticodons, such as *Neochloris aquatica* (NC_029670.1), *Bracteacoccus giganteas* (NC_028586.1), *Tetradesmus obliquus* (NC_008101.1), *Floydiella terrestris* (NC_014346.1), *Schizomeris leibleinii* (NC_015645.1), and *Oedogonium cardiacum* (NC_011031.1) [45].

The plastome size, structure, and gene content are highly conserved among *Garcinia* species. There were 18 genes (12 protein-coding genes and 6 tRNAs) containing intron(s) in the plastomes of *Garcinia* species. Although introns are not protein coding, they play an important role in gene expression by regulating the rate of transcription, nuclear export, and stability of transcripts [46,47]. The loss of introns such as *rpl*2 and *rps*16 has been reported in the plant plastomes [48,49,50,51], but we did not find any evidence of this in the plastomes of *Garcinia* species.

We detected two mis-annotations of genes (*infA* and *rpl32*) in *G. pedunculata* [20] and one mis-annotation of *infA* in *G. anomala* [52]. The *infA* is usually located between *rpl36* and *rps8*, whereas *rpl32* is usually located between *ndhF* and *trnL-UAG* [53]. Instead, the annotated *infA* was located within the *rpoC1* gene (*G. anomala*: MW582313 [52]; *G. pedunculata*: NC048983 [54]), while *rpl32* of *G. pedunculata* was located between *rpoB* and *trnC-GCA*. Hence, both genes were actually not found in the *Garcinia* plastomes, in agreement with a previous report that both *infA* and *rpl32* genes were lost in *G. mangostana* [55]. Plastid gene transfers to the nuclear genome (e.g., *accD*, *infA*, *rpl22*, and *rpl32*) have been documented in several plants [53,56,57]. The abundance of *infA* and *rpl32* transcripts in the seed transcriptome [4] suggests the same scenario for *G. mangostana*.

Genetic variations in *G. mangostana* cultivars have been shown by randomly amplified DNA fingerprinting (RAF) and inter simple sequence repeat (ISSR) molecular markers [58,59,60]. Plastomes also contain SSR and long repeats [61,62,63,64]. SSR, which is a stretch of 1–6 bp small repeats, is found extensively in different regions of the plastome, such as the intergenic regions, intron regions, and protein-coding regions [24]. In contrast, the long repeats found in the plastomes mostly fall within the intergenic region, although some of them were present in protein-coding genes [65]. Repetitive regions might lead to species variation as they have a higher tendency of recombination, translocation, and insertion/deletion [66]. In this study, SSR and long repeat analyses of plastomes showed variations among *Garcinia* species and varieties of *G. mangostana*. This supports the idea that both molecular markers are useful for species identification and taxonomic studies [67,68,69].

One of the main factors that contribute to the plastome size differences is the inverted repeat (IR) expansion and contraction [70,71]. For instance, nine genes were transferred from the SSC to the IR region in *Plantago ovata*, resulting in an extremely long IR (37.4 kb) [26] as compared to IR found in the other plastomes, which normally range between 25 and 30 kb [42]. In contrast, the loss of IR had been reported in the plastomes of *Vicia bungei* [72] and 25 durian varieties recently [38]. Besides long inversion, small-to-medium-sized (<few hundred base pairs) inversion in the plastome was also commonly found in angiosperm. Inversion between *trnV-UAC* and the *atpE* gene had been reported for the first time in the plastome of the Thai variety [15] and it is also found in the plastomes of all other seven *Garcinia* species used in this study. Small inversion in the plastome was also reported in *P. maritima* [26], *Panax schinseng* [73], Urticaceae family [51], and *Lindera* species [74]. Generally, this inversion is flanked by palindromes or quasi-palindrome (8–50 bp) to form the hairpin loop, and it is suggested that small inversion occurrence was affected by hairpin thermodynamic stability [75].

The phylogenomic analysis showed identical protein-coding genes between Mesta and Manggis varieties, which implies the same maternal lineage for both varieties from Malaysia compared to the Thailand variety (Figure 7). This result is congruent with the analysis using the whole plastome sequences of 16 species (Appendix A). The Malaysian (Mesta/Manggis) and Thailand mangosteen varieties did not cluster together in the phylogenomic analysis, contrary to the initial hypothesis of this study, which assumed a close phylogenomic relationship of the same species. Analysis of polymorphic sites of the 74 CDS used for phylogenomic tree construction showed a total of 559 variable sites, accounting for 0.85% differences between Manggis/Mesta and Thailand varieties (Appendix A). This was inconsistent with the clustering of different *G. mangostana* varieties, including the Thailand varieties in a previous phylogenetic study based on the nuclear ITS sequences [10].

To further investigate this discrepancy, we obtained the consensus ITS sequences of both Mesta (accession number: OK576276) and Manggis (accession number: OK576274) varieties by mapping the respective Illumina filtered reads against the published ITS sequence (accession number: AF367215). We reconstructed a phylogenetic tree [10] on the basis of the ITS sequences of other *Garcinia* species found in GenBank, including *G. celebica*, *G. gummi-gutta*, *G. hombroniana*, *G. oblongifolia*, *G. paucinervis*, and *G. pedunculata* (Appendix A). The ITS phylogenetic tree (Appendix A) showed that all the *G. mangostana* varieties were grouped with *G. malaccensis*, congruent with the results of the previous study [10]. Meanwhile, Mesta and Manggis were distantly related to *G. gummi-gutta*, *G. oblongifolia*, *G. paucinervis*, and *G. pedunculata*. Furthermore, Mesta, Masta, and *G. malaccensis* MY4 were clustered together, away from the Manggis variety. This indicates genetic differences between the two varieties despite near identical plastomes and supports that both varieties might have originated from *G. malaccensis*.

We found that both Manggis and Mesta varieties showed heterozygosity at certain positions of the ITS (Manggis: position 200; Mesta: position 444, 477, and 527) according to Illumina short reads results (Appendix A, respectively). Out of ten *G. mangostana* reported in the previous study [10], only one sample (*G. mangostana* TH3) from Thailand showed heterozygosity. Hence, *G. mangostana* may not have been derived from the hybridization of *G. hombroniana* and *G. malaccensis* [9]. Meanwhile, near-identical Mesta and Manggis plastomes (Figure 1 and Figure 6) indicate the same maternal lineage. This means the different evolutionary inferences from the nuclear genome and plastome analysis as plastids are inherited maternally compared to recombination events in nuclear genomes during reproduction [12].

As hybridization is a common practice in plant breeding to produce hybrids with desirable traits [76], the genetic variations and heterozygosity observed in this study could be due to the different germplasms. Different germplasms may hybridize via selective breeding and could have produced different varieties of *G. mangostana* in Malaysia and Thailand [10]. However, this remains highly speculative and requires further investigations of mangosteen from different biogeographical origins as well as plastomes of *G. celebica* (syn. *G. hombroniana*), *G. malaccensis*, *G. penangiana*, and *G. opaca* to ascertain their maternal lineages.

## 4. Materials and Methods

### 4.1. Mesta Plastome De Novo Genome Assembly

Genome sequences of the Mesta variety were obtained from the NCBI SRA database with the accession numbers SRX2718652 to SRX2718659 for PacBio long-read data (9.5 Gb) [17] and SRX270978 for Illumina short reads (50.2 Gb) [18]. CANU v2.0 [77] was used to perform PacBio raw data correction, trimming, and assembly using default parameters with minor modifications (useGrid = false, genomeSize = 6 g, batMemory = 252, batThreads = 32, minInputCoverage = 0.15, stopOnLowCoverage = 0). The draft genome assembly was polished with Illumina data using Pilon v1.23 [77]. Candidate plastome contigs were identified by using the BLAT v36.0 alignment tool with the previously reported *G. mangostana* (NC_036341.1) as the query. The identified contig was manually curated on the basis of the read coverage to obtain the final plastome of Mesta for subsequent analysis.

### 4.2. Manggis Plastome Assembly

Genome sequences of the Manggis variety were obtained from the NCBI SRA database with the accession number SRX1426419 for Illumina reads (51.1 Gb) [16]. Different methods were used for Manggis variety plastome assembly: (1) reference-guided assembly using GetOrganelle v1.7.5 [78], (2) de novo assembly using GetOrganelle v1.7.5 [78], and (3) de novo assembly using Platanus v1.2.4 [79] (Appendix A). To select the reference for reference-guided genome assembly, Manggis clean reads were aligned against the complete plastomes of Mesta and Thailand [15] varieties using bwa-mem v0.7.17 [80] and samtools v1.1 [81]. Next, the mapping coverage was visualized using weeSAM v1.6 (https://github.com/centre-for-virus-research/weeSAM; accessed on 24 December 2020). The reference with higher percentage coverage was chosen as the final reference for subsequent analysis. Manual curation was performed on the reference-guided assembled Manggis plastome to obtain the final Manggis plastome (Appendix A). The complete plastome sequences of *Garcinia* mangostana varieties Mesta and Manggis have been submitted to GenBank (https://www.ncbi.nlm.nih.gov/nuccore/; accessed on 21 April 2022) with the accession numbers MZ823408 and OK572535, respectively.

### 4.3. Plastome Annotation

Plastome annotation was performed online using GeSeq (https://chlorobox.mpimp-golm.mpg.de/geseq.html; accessed on 28 December 2020) [82]. Four *Garcinia* species (*Garcinia gummi-gutta* (NC_047250); *Garcinia mangostana* (NC_036341); *Garcinia oblongifolia* (NC_050384); and *Garcinia pedunculata* (NC_048983)) were used as BLAST-like Alignment Tool (BLAT) references. Respective gene annotations were corrected manually. Lastly, the plastome map was generated using the Organellar Genome DRAW (OGDRAW v1.3.1) program with default parameters [83]. The annotated plastomes of both Mesta and Manggis varieties were submitted to NCBI with accession numbers MZ823408 and OK572535, respectively.

### 4.4. Open Reading Frame (ORF) Coordinate Adjustment

The length of each gene found in *Garcinia* species was compared. Gene alignment was performed to visualize the differences when dissimilarity in gene length was detected by different annotation software. Next, manual coordinate adjustment was performed to standardize the 5′ end and the splicing site of these genes (Appendix A). The adjusted OFR coordinates (Appendix A) were used for subsequent analysis.

### 4.5. Identification of Simple-Sequence Repeats (SSRs)

The MISA-web microsatellite identification tool v2.1 (https://webblast.ipk-gatersleben.de/misa/; accessed on 6 February 2021) [84] was used to identify SSRs with the following default parameters: the minimum number of repeats for SSR motif of mono-, di-, tri-, tetra-, penta-, and hexa- were set to 10, 6, 5, 5, 5, and 5, respectively; the maximum length of the sequence between two SSRs to be registered as a compound SSR was set as 100 bp.

### 4.6. Long Repeat Analysis

Web-based REPuter (https://bibiserv.cebitec.uni-bielefeld.de/reputer/; accessed on 9 March 2021) [85] was used to identify forward, reverse, complement, and palindromic repeat sequences using the default setting of 50 for maximum computed repeats; hamming distance was set to 3, and minimal repeat size was set to 30 bp [85].

### 4.7. Codon Usage Analysis

Codon usage and relative synonymous codon usage (RSCU) value of all the annotated protein-coding genes presented in the plastomes of *Garcinia* species were analyzed using the MEGA X software v10.2.1 [86]. The RSCU with value >1.00 refers to a codon that is frequently used, whereas RSCU with the value <1.00 refers to a codon that is less frequently used. There is no codon usage bias when the RSCU value = 1.00 [87].

### 4.8. Plastome Sequence Alignment and Comparative Analysis

Plastome alignment and visualization were performed using the online comparison tool mVista (https://genome.lbl.gov/vista/mvista/submit.shtml; accessed on 27 May 2021) in LAGAN mode [88,89]. Mesta was used as a reference for alignment. The inverted repeat (IR) regions and the junction sites of the large single-copy (LSC) and small single-copy (SSC) regions of all the *Garcinia* species were compared using the IRscope online webtool [90] for the visualization of the expansion or contraction events. For both analyses, *Erythroxylum novogranatense*, from the Erythroxylaceae family of the same order Malpighiales, was included.

Mauve v.2.4.0 with progressiveMauve [91] was used to detect plastome inversions using default settings. A total of 17 species plastomes from the order Malpighiales (Appendix A) were aligned against *E. novogranatense* as a reference plastome. Palindrome in Galaxy Europe version 5.0.0.1 was used to detect palindromes with minimum and maximum length of palindromes set to 15 and 50 each, and the maximum gap between repeated regions was set to 400 bp.

### 4.9. Phylogenomic Analysis

For phylogenomic analysis, sixteen species were used: six *Garcinia* species (including *G. mangostana* var. Manggis, Mesta, and Thailand), five *Populus* species, two *Viola* species, *Erythroxylum novogranatense*, and *Jatropha curcas* from the order Malpighiales and *Arabidopsis thaliana* from the order Brassicales. A total of 74 protein-coding genes (Appendix A) that are found in all plastomes of these 16 species (including three varieties from *G. mangsotana*) were downloaded from the NCBI Organelle Genome database. These protein-coding genes were concatenated before being aligned using the MAFFT version 7 online tool (https://mafft.cbrc.jp/alignment/server/; accessed on 4 April 2021) [92]. Next, ModelTest-NG v0.1.6 [93] was used to select the DNA Evolutionary Models. The best model selected was GTR + I + G4, and it was used in the subsequent maximum likelihood (ML) analysis using the RAxML-NG v1.0.2 tool [94] with 1000 bootstrap replicates.

## 5. Conclusions

The complete plastomes of both Mesta and Manggis varieties of *G. mangostana* from Malaysia were successfully assembled and analyzed. PacBio long-read sequencing data helped to resolve the repetitive sequences in Mesta. Subsequently, this allowed for reference-guided genome assembly of the Manggis plastome. Notably, the Manggis plastome was almost identical with the Mesta plastome compared to the plastome of the Thailand variety. Comparative analysis showed that the gene structure, gene content, gene order, and gene orientation of *Garcinia* plastomes were largely conserved, except for one missing *trnH-GUG* gene in *G. gummi-gutta*. Phylogenomic analysis indicated that the Mesta and Manggis varieties were closer to *G. anomala*, *G. gummi-gutta*, and *G. oblongifolia*, while the Thailand variety clustered with *G. pedunculata*. Phylogenetic analysis based on the nuclear ITS sequences separated the Mesta and Manggis varieties on the basis of differences in their nuclear genomes. This study suggests different origins of the Mesta/Manggis and Thailand varieties. SSR and long repeats of plastomes identified in this study will provide useful biomarkers for species/variety identification and future lineage study of *Garcinia* genus.

## Figures and Tables

**Figure 1 plants-12-00930-f001:**
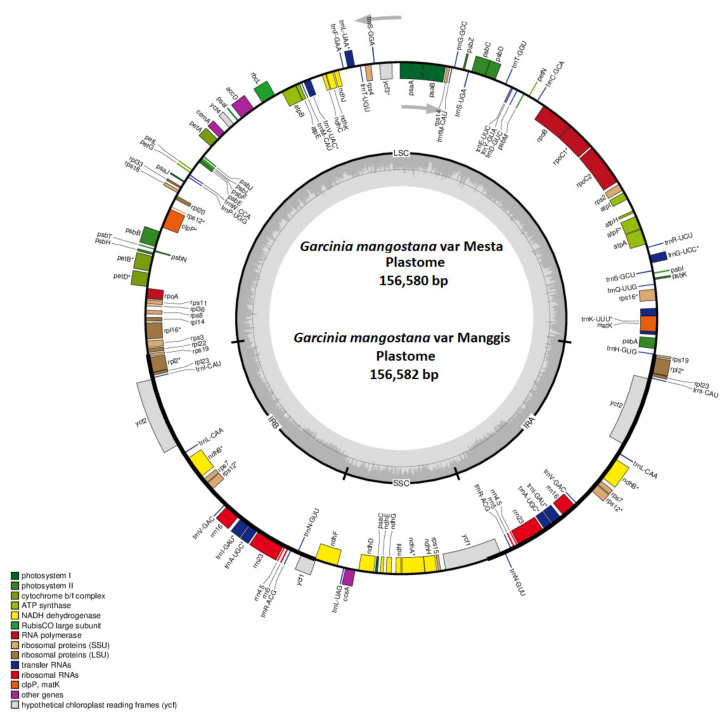
The circular plastome of the *G. mangostana* variety Mesta and Manggis. Genes inside the circle are transcribed clockwise while genes outside the circle are transcribed anti-clockwise, as indicated by the gray arrows. The gray bars inside the circle represent the GC content of the sequence. Asterisks (*) indicate genes containing intron(s).

**Figure 2 plants-12-00930-f002:**
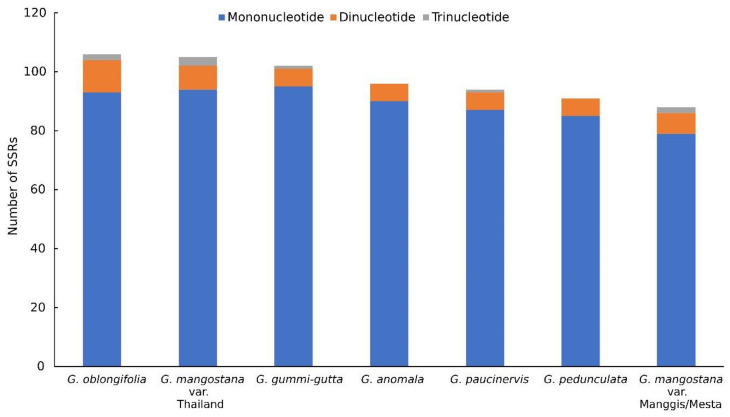
SSR analysis of six *Garcinia* species plastomes.

**Figure 3 plants-12-00930-f003:**
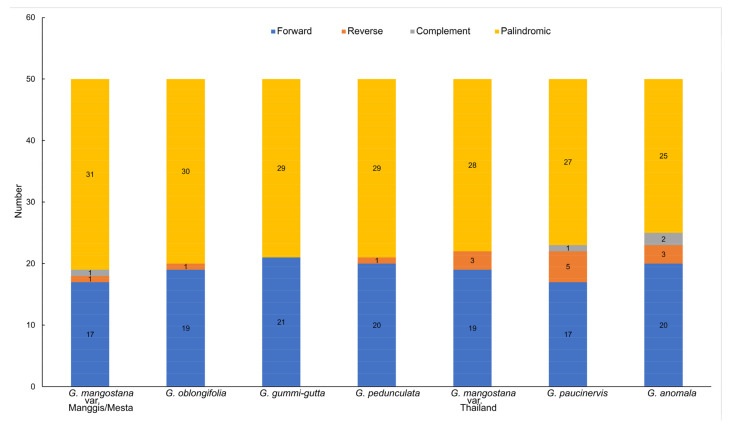
Long repeat analysis of six *Garcinia* species plastomes.

**Figure 4 plants-12-00930-f004:**
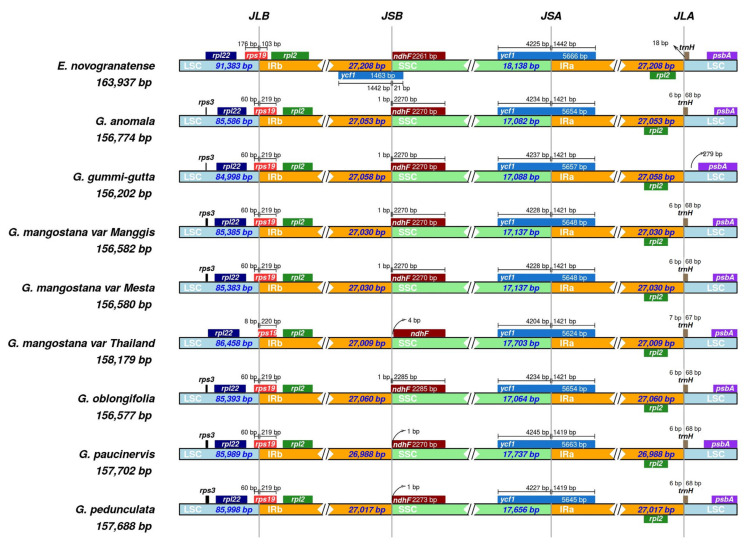
Comparison of genes on the borders of the LSC, SSC, and IR regions among six *Garcinia* plastomes and *Erythroxylum novogranatense*. Corresponding plastome size is shown on the left of each track. The intervals show the distance between the start and end coordinates of a particular gene from the junction sites, namely, JLB (LSC/IRb), JSB (SSC/IRb), JSA (SSC/IRa), and JLA (LSC/IRa). The sequence length in each region is annotated for genes spanning the junction sites.

**Figure 5 plants-12-00930-f005:**
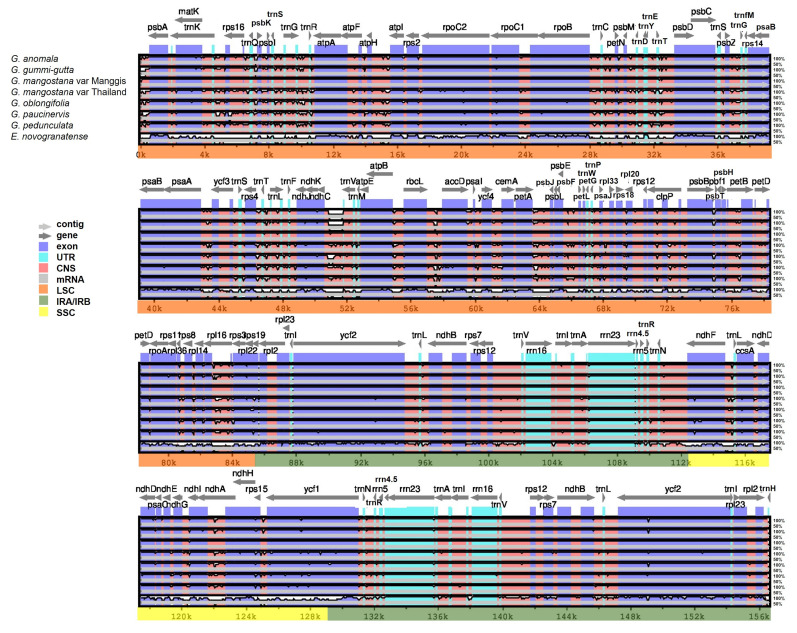
Alignment visualization of *Garcinia* species using Mesta as a reference by using the mVista alignment program. CNS: conserved non-coding sequences; UTR: untranslated region. The gray arrows above the alignment indicate the direction of the gene transcription. The identity percentage (50–100%) was indicated at the right-side of the mVista plot.

**Figure 6 plants-12-00930-f006:**
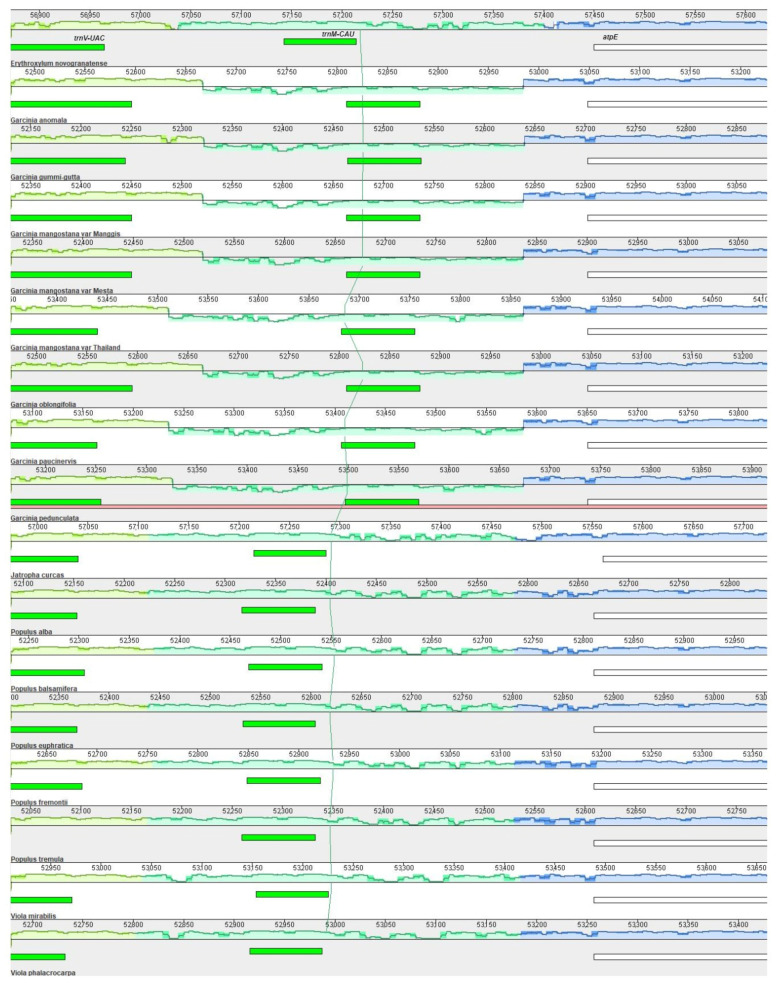
Multiple alignment using *E. novogranatense* as a reference. Color bars indicate syntenic blocks, and the connecting line indicates the correspondence of blocks across plastomes. There was a small inversion (jade-colored region below the x-axis) between *trnV-UAC* and the *atpE* gene (indicated above the green and white horizontal bars, respectively) shared by all *Garcinia* species.

**Figure 7 plants-12-00930-f007:**
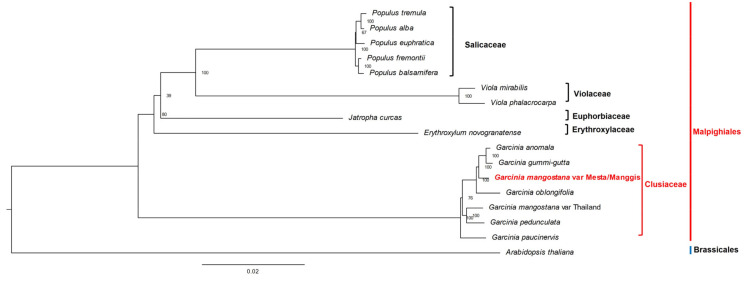
Phylogenetic tree (maximum likelihood) construction of 16 species (three varieties from *G. mangostana*) based on 74 protein-coding genes. The red outer line indicates order name, while the inner line indicates the family name.

**Table 1 plants-12-00930-t001:** Summary statistics of plastomes from different *Garcinia* species.

Species		Plastome Size (bp)	Size (bp)			Number of Genes *				GC Content (%)			
			LSC	SSC	IR	All	Protein-Coding	rRNA	tRNA	All	LSC	SSC	IR
*G. anomala*		156,774	85,586	17,082	27,053	128 (111)	83 (77)	8 (4)	37 (30)	36.1	33.5	30.3	42.1
*G. gummi-gutta*		156,202	84,996	17,088	27,059	127 (110)	83 (77)	8 (4)	36 (29)	36.2	33.5	30.3	42.1
*G. mangostana*	Manggis	156,582	85,385	17,137	27,030	128 (111)	83 (77)	8 (4)	37 (30)	36.2	33.6	30.2	42.2
	Mesta	156,580	85,383	17,137	27,030	128 (111)	83 (77)	8 (4)	37 (30)	36.2	33.6	30.2	42.2
	Thailand	158,179	86,458	17,703	27,009	128 (111)	83 (77)	8 (4)	37 (30)	36.1	33.5	30.1	42.2
*G. oblongifolia*		156,577	85,393	17,064	27,060	128 (111)	83 (77)	8 (4)	37 (30)	36.2	33.6	30.3	42.2
*G. paucinervis*		157,702	85,989	17,737	26,988	128 (111)	83 (77)	8 (4)	37 (30)	36.2	33.6	30.3	42.2
*G. pedunculata*		157,688	85,998	17,656	27,017	128 (111)	83 (77)	8 (4)	37 (30)	36.2	33.6	30.2	42.2

* Parentheses indicate the number of unique genes.

**Table 2 plants-12-00930-t002:** List of annotated genes in plastomes of Mesta and Manggis.

Function	Group	Gene Name
Protein synthesis and DNA replication	Transfer RNA	*trnA-UGC* (2×), *trnC-GCA*, *trnD-GUC*, *trnE-UUC*, *trnF-GAA*, *trnfM-CAU*, *trnG-GCC*, *trnH-GUG*, *trnK-UUU*, *trnI-GAU* (2×), *trnL-CAA* (2×), *trnL-UAA*, *trnL-UAG*, *trnM-CAU*, *trnI-CAU* (2×), *trnN-GUU* (2×), *trnP-UGG*, *trnQ-UUG*, *trnR-ACG* (2×), *trnR-UCU*, *trnS-CGA*, *trnS-GCU*, *trnS-GGA*, *trnS-UGA*, *trnT-GGU*, *trnT-UGU*, *trnV-GAC* (2×), *trnV-UAC*, *trnW-CCA*, *trnY-GUA*
	Ribosomal RNA	*rrn16* (2×), *rrn23* (2×), *rrn4.5* (2×), *rrn5* (2×)
	Ribosomal protein small subunit	*rps2*, *rps3*, *rps4*, *rps7* (2×), *rps8*, *rps11*, *rps12* (2×), *rps14*, *rps15*, *rps16*, *rps18*, *rps19*
	Ribosomal protein large subunit	*rpl2* (2×), *rpl14*, *rpl16*, *rpl20*, *rpl22*, *rpl23* (2×), *rpl33*, *rpl36*
	Subunits of RNA polymerase	*rpoA*, *rpoB*, *rpoC1*, *rpoC2*
Photosynthesis	Photosystem I	*psaA*, *psaB*, *psaC*, *psaI*, *psaJ*
	Photosystem II	*psbA*, *psbB*, *psbC*, *psbD*, *psbE*, *psbF*, *psbH*, *psbI*, *psbJ*, *psbK*, *psbL*, *psbM*, *psbT*, *psbZ*
	Cytochrome b_6_f complex	*petA*, *petB*, *petD*, *petG*, *petL*, *petN*
	ATP synthase	*atpA*, *atpB*, *atpE*, *atpF*, *atpH*, *atpI*
	NADH-dehydrogenase	*ndhA*, *ndhB* (2×), *ndhC*, *ndhD*, *ndhE*, *ndhF*, *ndhG*, *ndhH*, *ndhI*, *ndhJ*, *ndhK*
	Large subunit Rubisco	*rbcL*
Miscellaneous group	Photosystem II protein N	*psbN*
	Acetyl-CoA carboxylase	*accD*
	Cytochrome c biogenesis	*ccsA*
	Maturase	*matK*
	ATP-dependent protease	*clpP*
	Inner membrane protein	*cemA*
Pseudogene unknown function	Conserved hypothetical chloroplast ORF	*ycf1*, *ycf2* (2×), *ycf3*, *ycf4*

## Data Availability

The complete plastome sequences and ITS sequences of *Garcinia mangostana* var. Mesta and Manggis can be accessed via GenBank (https://www.ncbi.nlm.nih.gov/nuccore/, accessed on 12 October 2022).

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
