# Peer review of "Plastomes of Garcinia mangostana L. and Comparative Analysis with Other Garcinia Species"

_plants, 2023, doi:10.3390/plants12040930_

Round 1
Reviewer 1 Report
The manuscript entitled “Plastomes of Garcinia mangostana L. and comparative analysis with other Garcinia species” is of great interest for plant scientist community. Although the manuscript contains good information but there are some flaws that must be addressed and fulfilled for the validation of study and outputs. For improvement of manuscript, consider the following suggestions and comments.
Introduction: First and second paragraph must be merged to have better understanding of mangosteen geographic origin and importance.
References about first discovery of genes must be added in the Table 2.
Materials section is well documented.
Discussion: This section is very well written; however, few more recent citations should be included to make it worth reading.
Author Response
Reviewer 1
1. Introduction: First and second paragraph must be merged to have better understanding of mangosteen geographic origin and importance.
First and second paragraph were merged. Line 30-48
2. References about first discovery of genes must be added in the Table 2.
Table 2 aimed to list down the genes in plastomes of Mesta and Manggis. We do not think adding the references about the first discovery of genes is needed in this case as in many of the previous publications cited in the references.
3. Discussion: This section is very well written; however, few more recent citations should be included to make it worth reading.
Recent citations were included:
Here, the use of long reads for the assembly of the plastid genomes was ideal to obtain longer contigs and to resolve repetitive regions [1-4]. (line 281-283)
The loss of introns such as rpl2 and rps16 has been reported in the plant plastomes [5-8] but, we did not find any evidence of this in the plastomes of Garcinia species. (line 303-305)
Plastomes also contain SSR and long repeats [9-12]. (line 318)
In contrast, the loss of IR had been reported in the plastomes of Vicia bungei [13] and 25 durian varieties recently [3]. (line 331-332)
Small inversion in plastome was also reported in P. maritima [14], Panax schinseng [15], Urticeae [8], and Lindera species [16]. (line 336-337).
References
- Chen, X.; Li, Q.; Li, Y.; Qian, J.; Han, J. Chloroplast genome of Aconitum barbatum var. puberulum (Ranunculaceae) derived from CCS reads using the PacBio RS platform. Frontiers in Plant Science 2015, 6, 42.
- Ferrarini, M.; Moretto, M.; Ward, J.A.; Šurbanovski, N.; Stevanović, V.; Giongo, L.; Viola, R.; Cavalieri, D.; Velasco, R.; Cestaro, A. An evaluation of the PacBio RS platform for sequencing and de novo assembly of a chloroplast genome. BMC genomics 2013, 14, 1-12.
- Shearman, J.R.; Sonthirod, C.; Naktang, C.; Sangsrakru, D.; Yoocha, T.; Chatbanyong, R.; Vorakuldumrongchai, S.; Chusri, O.; Tangphatsornruang, S.; Pootakham, W. Assembly of the durian chloroplast genome using long PacBio reads. Scientific reports 2020, 10, 1-8.
- Fahrenkrog, A.M.; Matsumoto, G.; Toth, K.; Jokipii-Lukkari, S.; Salo, H.M.; Häggman, H.; Benevenuto, J.; Munoz, P. Chloroplast genome assemblies and comparative analyses of major <em>Vaccinium</em> berry crops. bioRxiv 2022, 2022.2002.2023.481500, doi:10.1101/2022.02.23.481500.
- Liu, L.; Wang, Y.; He, P.; Li, P.; Lee, J.; Soltis, D.E.; Fu, C. Chloroplast genome analyses and genomic resource development for epilithic sister genera Oresitrophe and Mukdenia (Saxifragaceae), using genome skimming data. BMC Genomics 2018, 19, 235, doi:10.1186/s12864-018-4633-x.
- Downie, S.R.; Olmstead, R.G.; Zurawski, G.; Soltis, D.E.; Soltis, P.S.; Watson, J.C.; Palmer, J.D. Six independent losses of the chloroplast DNA rpl2 intron in dicotyledons: molecular and phylogenetic implications. Evolution 1991, 45, 1245-1259.
- Haberle, R.C.; Fourcade, H.M.; Boore, J.L.; Jansen, R.K. Extensive rearrangements in the chloroplast genome of Trachelium caeruleum are associated with repeats and tRNA genes. Journal of Molecular Evolution 2008, 66, 350-361.
- Ogoma, C.A.; Liu, J.; Stull, G.W.; Wambulwa, M.C.; Oyebanji, O.; Milne, R.I.; Monro, A.K.; Zhao, Y.; Li, D.-Z.; Wu, Z.-Y. Deep insights into the plastome evolution and phylogenetic relationships of the tribe Urticeae (Family urticaceae). Frontiers in plant science 2022, 13.
- Zhu, M.; Feng, P.; Ping, J.; Li, J.; Su, Y.; Wang, T. Phylogenetic significance of the characteristics of simple sequence repeats at the genus level based on the complete chloroplast genome sequences of Cyatheaceae. Ecology and evolution 2021, 11, 14327-14340.
- Alzahrani, D.A.; Yaradua, S.S.; Albokhari, E.J.; Abba, A. Complete chloroplast genome sequence of Barleria prionitis, comparative chloroplast genomics and phylogenetic relationships among Acanthoideae. BMC Genomics 2020, 21, 393, doi:10.1186/s12864-020-06798-2.
- Asaf, S.; Ahmad, W.; Al-Harrasi, A.; Khan, A.L. Uncovering the first complete plastome genomics, comparative analyses, and phylogenetic dispositions of endemic medicinal plant Ziziphus hajarensis (Rhamnaceae). BMC Genomics 2022, 23, 83, doi:10.1186/s12864-022-08320-2.
- Zhan, X.; Zhang, Z.; Zhang, Y.; Gao, Y.; Jin, Y.; Shen, C.; Wang, H.; Feng, S. Complete Plastome of Physalis angulata var. villosa, Gene Organization, Comparative Genomics and Phylogenetic Relationships among Solanaceae. Genes 2022, 13, 2291.
- Jo, I.-H.; Han, S.; Shim, D.; Ryu, H.; Hyun, T.K.; Lee, Y.; Kim, D.; So, Y.-S.; Chung, J.-W. Complete Chloroplast Genome of the Inverted Repeat-Lacking Species Vicia bungei and Development of Polymorphic Simple Sequence Repeat Markers. Frontiers in Plant Science 2022, 1571.
- Asaf, S.; Khan, A.L.; Khan, A.; Khan, G.; Lee, I.-J.; Al-Harrasi, A. Expanded inverted repeat region with large scale inversion in the first complete plastid genome sequence of Plantago ovata. Scientific reports 2020, 10, 1-16.
- Kim, K.-J.; Lee, H.-L. Complete chloroplast genome sequences from Korean ginseng (Panax schinseng Nees) and comparative analysis of sequence evolution among 17 vascular plants. DNA research 2004, 11, 247-261.
- Jo, S.; Kim, Y.-K.; Cheon, S.-H.; Fan, Q.; Kim, K.-J. Characterization of 20 complete plastomes from the tribe Laureae (Lauraceae) and distribution of small inversions. Plos one 2019, 14, e0224622.
Reviewer 2 Report
In this manuscript, Wee et al. report the plastid genome assemblies of two Garcinia species and clarify their phylogenetic positions in this genus. Authors performed assembly based on the combination of PacBio long-read and Illumina short-read and made meticulous annotations for such two plastid genomes. Comparative genomic and phylogenomic analysis based on the plastid genome assemblies of related species was also well conducted. The manuscript is logically organized and well-written. Overall, I think the very clear analyses presented in this manuscript provide a useful genomic resource for further phylogeny cognition of Garcinia species. I have some minor issues hope the authors may consider:
Line 123. What is the meaning of 18 unique genes? Garcinia- plastid-specific gene?
Lines 135-137. Meaning of the numbers here is not clear.
Line 163. SSR no. 3.
Line 165. “between two SRRs” is not clear to me.
Line 234-235. It’s better to mark the regions of LSC, SSC, and IR in Figure 5. Also, the two conclusions from Figure 5 are not immediately clear and it will help to give some statistical evidence.
I don’t understand Figure 6. How to judge inversion from this figure? The region below the x-axis? Please add more detailed information in the figure legend.
Line 403. Citation of weeSam.
Line 469. How did the author determine the common genes (orthologs) among the 16 species?
Line 473-475. How many bootstrap replicates were performed when constructing the phylogeny tree?
Line 487-488. Meaning is not clear. Please rephrase.
Author Response
Reviewer 2
Line 123. What is the meaning of 18 unique genes? Garcinia- plastid-specific gene?
Unique in this context means single-copy non-redundant genes. We have rephrased (line 123-127) for clarity:
In the plastomes of six Garcinia species (including G. mangostana var. Manggis, Mesta, and Thailand) used in this comparative analysis, there were a total of 18 single-copy non-redundant plastid genes (rps16, atpF, rpoC1, ycf3, rps12, clpP, petB, petD, rpl16, rpl2, ndhB, ndhA, trnK-UUU, trnG-UCC, trnL-UAA, trnV-UAC, trnI-GAU, and trnA-UGC) containing at least one intron with two introns in clpP and rps12 (Supplementary Table S5), which is similar to those generally found in other plants.
Lines 135-137. Meaning of the numbers here is not clear.
The numbers here refer to the total number of codon usage as clarified in Line 136-140:
The total number of codon usage in 83 protein-coding genes found in the plastomes was different among the Garcinia species ranging from 26,195 in G. pedunculata, 26,216 in Thailand variety, 26,244 in G. paucinervies, 26,249 in G. anomala, 26,257 in both Manggis and Mesta varieties, 26,265 in G. gummi-gutta, to 26,268 in G. oblongifolia (refer to ‘total number of codon usage’ in Supplementary Table S6).
Line 163. SSR no. 3.
Line 164-168
The SSRs identified in the plastomes of both Mesta and Manggis were nearly the same, except Manggis had additional mononucleotides, A [SSR (A)14 and (A)15 at no. 3] and C [SSR (C)12(A)12 and (C)13(A)12 at no. 25], as per Supplementary Table S7 and Supplementary Table S8, respectively for Mesta and Manggis.
Line 165. “between two SRRs” is not clear to me.
Rephrased 168
There were 14 compound SSRs in both Mesta and Manggis.
Rephrased 438-439
The maximum length of the sequence between two SSRs to be registered as a compound SSR was set to be 100 bp.
Line 234-235. It’s better to mark the regions of LSC, SSC, and IR in Figure 5. Also, the two conclusions from Figure 5 are not immediately clear and it will help to give some statistical evidence.
Line 255
LSC, SSC, and IR regions were added in Figure 5.
Figure 5 shows the plastome comparison of six Garcinia species used in this study using Mesta as a reference. The diagram illustrated the qualitative differences among the plastomes. As such, there is no statistical analysis. The legend stated the region of LSC, SSC, IR, and exon (coding regions). Hence, the figure clearly showed that (1) IR regions were more conserved (higher % identity) compared with LSC and SSC regions, (2) coding regions were more conserved (lower % identity) than intergenic regions.
I don’t understand Figure 6. How to judge inversion from this figure? The region below the x-axis? Please add more detailed information in the figure legend.
More detailed information was added at lines 262-263
Figure 6. Multiple alignment using E. novogranatense as reference showed that there was a small inversion (pink region below the x-axis) found in Garcinia species between trnV-UAC (green bar on the left) and atpE gene (white bar on the right).
Line 403. Citation of weeSam.
Line 408
WeeSAM v.1.6 (https://github.com/centre-for-virus-research/weeSAM)
Line 469. How did the author determine the common genes (orthologs) among the 16 species?
Protein coding genes in the plastome of each species were downloaded from NCBI Organelle Genome database and compiled into Supplementary Table S10. The accession number for each species is provided in Supplementary Table S9. From there, genes that are common in all 16 species were used for phylogenomic analysis.
This is stated in Materials and Methods:
A total of 74 protein-coding genes (Supplementary Table S10) that are found in all plastomes of these 16 species (including three varieties from G. mangsotana) were downloaded from the NCBI Organelle Genome database. (line 473-475)
Line 473-475. How many bootstrap replicates were performed when constructing the phylogeny tree?
Line 477-479
The best model selected was GTR+I+G4 and it was used in the subsequent maximum likelihood (ML) analysis using RAxML-NG v1.0.2 tool [1] with 1,000 bootstrap replicates.
Line 487-488. Meaning is not clear. Please rephrase.
Line 490-492
Phylogenetic analysis based on the nuclear ITS sequences separated the Mesta and Manggis varieties based on differences in their nuclear genomes.
Reference
- Kozlov, A.M.; Darriba, D.; Flouri, T.; Morel, B.; Stamatakis, A. RAxML-NG: a fast, scalable and user-friendly tool for maximum likelihood phylogenetic inference. Bioinformatics 2019, 35, 4453-4455, doi:10.1093/bioinformatics/btz305.